# Lipschitz regularity of deep neural networks: analysis and efficient estimation

**Kevin Scaman**
Huawei Noah's Ark Lab
kevin.scaman@huawei.com

**Aladin Virmaux**
Huawei Noah's Ark Lab
aladin.virmaux@huawei.com

## Abstract

Deep neural networks are notorious for being sensitive to small well-chosen perturbations, and estimating the regularity of such architectures is of utmost importance for safe and robust practical applications. In this paper, we investigate one of the key characteristics to assess the regularity of such methods: the Lipschitz constant of deep learning architectures. First, we show that, even for two layer neural networks, the exact computation of this quantity is NP-hard and state-of-art methods may significantly overestimate it. Then, we both extend and improve previous estimation methods by providing *AutoLip*, the first generic algorithm for upper bounding the Lipschitz constant of any automatically differentiable function. We provide a power method algorithm working with automatic differentiation, allowing efficient computations even on large convolutions. Second, for sequential neural networks, we propose an improved algorithm named *SeqLip* that takes advantage of the linear computation graph to split the computation per pair of consecutive layers. Third we propose heuristics on *SeqLip* in order to tackle very large networks. Our experiments show that *SeqLip* can significantly improve on the existing upper bounds. Finally, we provide an implementation of *AutoLip* in the *PyTorch* environment that may be used to better estimate the robustness of a given neural network to small perturbations or regularize it using more precise Lipschitz estimations.

## 1 Introduction

Deep neural networks made a striking entree in machine learning and quickly became state-of-the-art algorithms in many tasks such as computer vision [1, 2, 3, 4], speech recognition and generation [5, 6] or natural language processing [7, 8].

However, deep neural networks are known for being very sensitive to their input, and *adversarial examples* provide a good illustration of their lack of robustness [9, 10]. Indeed, a well-chosen small perturbation of the input image can mislead a neural network and significantly decrease its classification accuracy. One metric to assess the robustness of neural networks to small perturbations is the *Lipschitz constant* (see Definition 1), which upper bounds the relationship between input perturbation and output variation for a given distance. For generative models, the recent *Wasserstein GAN* [11] improved the training stability of GANs by reformulating the optimization problem as a minimization of the Wasserstein distance between the real and generated distributions [12]. However, this method relies on an efficient way of constraining the Lipschitz constant of the critic, which was only partially addressed in the original paper, and the object of several follow-up works [13, 14].

Recently, Lipschitz continuity was used in order to improve the state-of-the-art in several deep learning topics: (1) for robust learning, avoiding adversarial attacks was achieved in [15] by constraining local Lipschitz constants in neural networks. (2) For generative models, using spectral normalization on each layer allowed [13] to successfully train a GAN on ILRSVRC2012 dataset. (3) In deep

learning theory, novel generalization bounds critically rely on the Lipschitz constant of the neural network [16, 17, 18].

To the best of our knowledge, the first upper-bound on the Lipschitz constant of a neural network was described in [9, Section 4.3], as the product of the spectral norms of linear layers (a special case of our generic algorithm, see Proposition 1). More recently, the Lipschitz constant of *scatter networks* was analyzed in [19]. Unfortunately, this analysis does not extend to more general architectures.

Our aim in this paper is to provide a rigorous and practice-oriented study on how Lipschitz constants of neural networks and automatically differentiable functions may be estimated. We first precisely define the notion of Lipschitz constant of vector valued functions in Section 2, and then show in Section 3 that its estimation is, even for 2-layer *Multi-Layer-Perceptrons* (MLP), **NP**-hard. In Section 4, we both extend and improve previous estimation methods by providing *AutoLip*, the first generic algorithm for upper bounding the Lipschitz constant of any automatically differentiable function. Moreover, we show how the Lipschitz constant of most neural network layers may be computed efficiently using automatic differentiation algorithms [20] and libraries such as PyTorch [21]. Notably, we extend the power method to convolution layers using automatic differentiation to speed-up the computations. In Section 6, we provide a theoretical analysis of AutoLip in the case of sequential neural networks, and show that the upper bound may lose a multiplicative factor *per activation layer*, which may significantly downgrade the estimation quality of AutoLip and lead to a very large and unrealistic upper bound. In order to prevent this, we propose an improved algorithm called *SeqLip* in the case of sequential neural networks, and show in Section 7 that SeqLip may significantly improve on AutoLip. Finally we discuss the different algorithms on the AlexNet [1] neural network for computer vision using the proposed algorithms. [1]

## 2   Background and notations

In the following, we denote as $\langle x, y \rangle$ and $\|x\|_2$ the scalar product and $L_2$-norm of the Hilbert space $\mathbb{R}^n$, $x \cdot y$ the coordinate-wise product of $x$ and $y$, and $f \circ g$ the composition between the functions $f : \mathbb{R}^k \to \mathbb{R}^m$ and $g : \mathbb{R}^n \to \mathbb{R}^k$. For any differentiable function $f : \mathbb{R}^n \to \mathbb{R}^m$ and any point $x \in \mathbb{R}^n$, we will denote as $\mathrm{D}_x f \in \mathbb{R}^{m \times n}$ the differential operator of $f$ at $x$, also called the *Jacobian matrix*. Note that, in the case of real valued functions (i.e. $m = 1$), the gradient of $f$ is the transpose of the differential operator: $\nabla f(x) = (\mathrm{D}_x f)^\top$. Finally, $\mathrm{diag}_{n,m}(x) \in \mathbb{R}^{n \times m}$ is the rectangular matrix with $x \in \mathbb{R}^{\min\{n,m\}}$ along the diagonal and 0 outside of it. When unambiguous, we will use the notation $\mathrm{diag}(x)$ instead of $\mathrm{diag}_{n,m}(x)$. All proofs are available as supplemental material.

**Definition 1.** A function $f : \mathbb{R}^n \to \mathbb{R}^m$ is called *Lipschitz continuous* if there exists a constant $L$ such that

$$\forall x, y \in \mathbb{R}^n, \ \|f(x) - f(y)\|_2 \le L \|x - y\|_2.$$

The smallest $L$ for which the previous inequality is true is called the *Lipschitz constant* of $f$ and will be denoted $L(f)$.

For locally Lipschitz functions (i.e. functions whose restriction to some neighborhood around any point is Lipschitz), the Lipschitz constant may be computed using its differential operator.

**Theorem 1** (Rademacher [22, Theorem 3.1.6]). *If $f : \mathbb{R}^n \to \mathbb{R}^m$ is a locally Lipschitz continuous function, then $f$ is differentiable almost everywhere. Moreover, if $f$ is Lipschitz continuous, then*

$$L(f) = \sup_{x \in \mathbb{R}^n} \| \mathrm{D}_x f \|_2 \tag{1}$$

*where $\|M\|_2 = \sup_{\{x \, : \, \|x\|=1\}} \|Mx\|_2$ is the operator norm of the matrix $M \in \mathbb{R}^{m \times n}$.*

In particular, if $f$ is real valued (i.e. $m = 1$), its Lipschitz constant is the maximum norm of its gradient $L(f) = \sup_x \|\nabla f(x)\|_2$ on its domain set. Note that the supremum in Theorem 1 is a slight abuse of notations, since the differential $\mathrm{D}_x f$ is defined *almost everywhere* in $\mathbb{R}^n$, except for a set of Lebesgue measure zero.

## 3 Exact Lipschitz computation is NP-hard

In this section, we show that the exact computation of the Lipschitz constant of neural networks is **NP**-hard, hence motivating the need for good approximation algorithms. More precisely, upper bounds are in this case more valuable as they ensure that the variation of the function, when subject to an input perturbation, remains small. A *neural network* is, in essence, a succession of linear operators and non-linear activation functions. The most simplistic model of neural network is the *Multi-Layer-Perceptron* (MLP) as defined below.

**Definition 2** (MLP). A $K$-layer *Multi-Layer-Perceptron* $f_{MLP} : \mathbb{R}^n \to \mathbb{R}^m$ is the function

$$f_{MLP}(x) = T_K \circ \rho_{K-1} \circ \cdots \circ \rho_1 \circ T_1(x),$$

where $T_k : x \mapsto M_k x + b_k$ is an affine function and $\rho_k : x \mapsto (g_k(x_i))_{i \in [\![1, n_k]\!]}$ is a non-linear activation function.

Many standard deep network architectures (*e.g.* CNNs) follow –to some extent– the MLP structure. It turns out that even for 2-layer MLPs, the computation of the Lipschitz constant is **NP**-hard.

**Problem 1** (**LIP-CST**). **LIP-CST** is the decision problem associated to the exact computation of the Lipschitz constant of a 2-layer MLP with ReLU activation layers.

> **Input:** Two matrices $M_1 \in \mathbb{R}^{l \times n}$ and $M_2 \in \mathbb{R}^{m \times l}$, and a constant $\ell \geq 0$.

> **Question:** Let $f = M_2 \circ \rho \circ M_1$ where $\rho(x) = \max\{0, x\}$ is the ReLU activation function. Is the Lipschitz constant $L(f) \leq \ell$ ?

Theorem 2 shows that, even for extremely simple neural networks, exact Lipschitz computation is not achievable in polynomial time (assuming that $\mathbf{P} \neq \mathbf{NP}$). The proof of Theorem 2 is available in the supplemental material.

**Theorem 2.** *Problem 1 is* **NP**-*hard.*

Theorem 2 relies on a reduction to the **NP**-hard problem of quadratic concave minimization on a hypercube by considering well-chosen matrices $M_1$ and $M_2$.

## 4 AutoLip: a Lipschitz upper bound through automatic differentiation

Efficient implementations of backpropagation in modern deep learning libraries such as *Py-Torch* [21] or *TensorFlow* [23] rely on on the concept of *automatic differentiation* [24, 20]. Simply put, automatic differentiation is a principled approach to the computation of gradients and differential operators of functions resulting from $K$ successive operations.

**Definition 3.** A function $f : \mathbb{R}^n \to \mathbb{R}^m$ is *computable in $K$ operations* if it is the result of $K$ *simple* functions in the following way: $\exists (\theta_1, ..., \theta_K)$ functions of the input $x$ and $(g_1, \ldots, g_K)$ where $g_l$ is a function of $(\theta_i)_{i \leq l-1}$ such that:

$$\theta_0(x) = x, \qquad \theta_K(x) = f(x), \qquad \forall k \in [\![1, K]\!], \; \theta_k(x) = g_k(x, \theta_1(x), \ldots, \theta_{k-1}(x)). \quad (2)$$

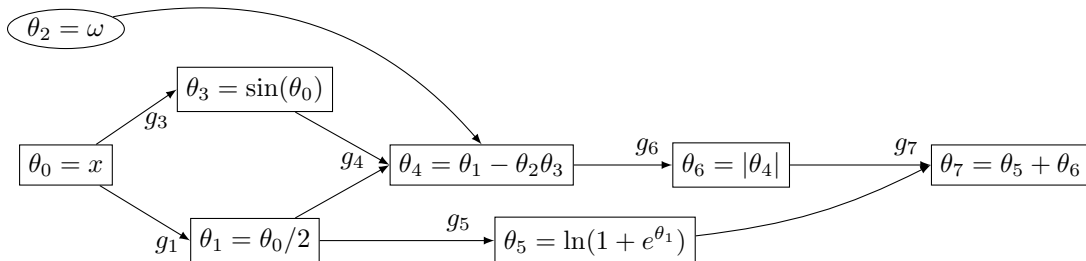

Figure 1: Example of a computation graph for $f_\omega(x) = \ln(1 + e^{x/2}) + |x/2 - \omega \sin(x)|$.

**Algorithm 1** AutoLip

---

**Input:** function $f : \mathbb{R}^n \to \mathbb{R}^m$ and its computation graph $(g_1, ..., g_K)$

**Output:** upper bound on the Lipschitz constant: $\hat{L}_{AL} \geq L(f)$

1:   $\mathcal{Z} = \{(z_0, ..., z_K) \; : \; \forall k \in [\![0, K]\!], \theta_k \text{ is constant} \Rightarrow z_k = \theta_k(0)\}$

2:   $L_0 \leftarrow 1$

3:   **for** $k = 1$ to $K$ **do**

4:      $L_k \leftarrow \sum\limits_{i=1}^{k-1} \max\limits_{z \in \mathcal{Z}} \|\partial_i g_k(z)\|_2 L_i$

5:   **end for**

6:   **return** $\hat{L}_{AL} = L_k$

---

We assume that these operations are all locally Lipschitz-continuous, and that their partial derivatives $\partial_i g_k(x)$ can be computed and efficiently maximized. This assumption is discussed in Section 5 for the main operations used in neural networks. When the function is real valued (i.e. $m = 1$), the backpropagation algorithm allows to compute its gradient efficiently in time proportional to the number of operations $K$ [25]. For the computation of the Lipschitz constant $L(f)$, a forward propagation through the computation graph is sufficient. More specifically, the chain rule immediately implies

$$\mathrm{D}_x \, \theta_k = \sum_{i=1}^{k-1} \partial_i g_k(\theta_0(x), \ldots, \theta_{k-1}(x)) \, \mathrm{D}_x \, \theta_i \,, \tag{3}$$

and taking the norm then maximizing over all possible values of $\theta_i(x)$ leads to the *AutoLip* algorithm described in Alg. (1). This algorithm is an extension of the well known product of operator norms for MLPs (see e.g. [13]) to any function computable in $K$ operations.

**Proposition 1.** *For any MLP (see Definition 2) with 1-Lipschitz activation functions (e.g. ReLU, Leaky ReLU, SoftPlus, Tanh, Sigmoid, ArcTan or Softsign), the AutoLip upper bound becomes*

$$\hat{L}_{AL} = \prod_{k=1}^{K} \|M_k\|_2.$$

Note that, when the intermediate function $\theta_k$ does not depend on $x$, it is not necessary to take a maximum over all possible values of $\theta_k(x)$. To this end we define the set of feasible intermediate values as

$$\mathcal{Z} = \{(z_0, ..., z_K) \; : \; \forall k \in [\![0, K]\!], \theta_k \text{ is constant} \Rightarrow z_k = \theta_k(0)\}, \tag{4}$$

and only maximize partial derivatives over this set. In practice, this is equivalent to removing branches of the computation graph that are not reachable from node 0 and replacing them by constant values. To illustrate this definition, consider a simple matrix product operation $f(x) = Wx$. One possible computation graph for $f$ is $\theta_0 = x$, $\theta_1 = W$ and $\theta_2 = g_2(\theta_0, \theta_1) = \theta_1 \theta_0$. While the quadratic function $g_2$ is not Lipschitz-continuous, its derivative w.r.t. $\theta_0$ is bounded by $\partial_0 g_2(\theta_0, \theta_1) = \theta_1 = W$. Since $\theta_1$ is constant relatively to $x$, we have $\mathcal{Z} = \{(x, 0)\}$ and the algorithm returns the exact Lipschitz constant $\hat{L}_{AL} = L(f) = \|W\|_2$.

**Example.** We consider the graph explicited on Figure 1. Since $\theta_2$ is a constant w.r.t. $x$, we can replace it by its value $\omega$ in all other nodes. Then, the AutoLip algorithm runs as follows:

$$\hat{L}_{AL} = L_7 = L_6 + L_5 = L_1 + L_4 = 2L_1 + wL_3 = 1 + \omega. \tag{5}$$

Note that, in this example, the Lipschitz upper bound $\hat{L}_{AL}$ matches the exact Lipschitz constant $L(f_\omega) = 1 + \omega$.

## 5   Lipschitz constants of typical neural network layers

**Linear and convolution layers.** The Lipschitz constant of an affine function $f : x \mapsto Mx + b$ where $M \in \mathbb{R}^{m \times n}$ and $b \in \mathbb{R}^m$ is the largest singular value of its associated matrix $M$, which may be computed efficiently, up to a given precision, using the *power method* [26]. In the case of

convolutions, the associated matrix may be difficult to access and high dimensional, hence making the direct use of the power method impractical. To circumvent this difficulty, we extend the power method to any affine function on whose automatic differentiation can be used (e.g. linear or convolution layers of neural networks) by noting that the only matrix multiplication of the power method $M^\top M x$ can be computed by differentiating a well-chosen function.

**Lemma 1.** *Let $M \in \mathbb{R}^{m \times n}$, $b \in \mathbb{R}^m$ and $f : x \mapsto Mx + b$ be an affine function. Then, for all $x \in \mathbb{R}^n$, we have*

$$M^\top M x \quad = \quad \nabla g(x),$$

*where $g(x) = \frac{1}{2}\|f(x) - f(0)\|_2^2$.*

*Proof.* By definition, $g(x) = \frac{1}{2}\|Mx\|_2^2$, and differentiating this equation leads to the desired result. $\square$

---

**Algorithm 2** AutoGrad compliant power method

---

**Input:** affine function $f : \mathbb{R}^n \to \mathbb{R}^m$, number of iteration $N$
**Output:** approximation of the Lipschitz constant $L(f)$
 1: **for** $k = 1$ to $N$ **do**
 2:     $v \leftarrow \nabla g(v)$ where $g(x) = \frac{1}{2}\|f(x) - f(0)\|_2^2$
 3:     $\lambda \leftarrow \|v\|_2$
 4:     $v \leftarrow v/\lambda$
 5: **end for**
 6: **return** $L(f) = \|f(v) - f(0)\|_2$

---

The full algorithm is described in Alg. (2). Note that this algorithm is fully compliant with any dynamic graph deep learning libraries such as PyTorch. The gradient of the square norm may be computed through autograd, and the gradient of $L(f)$ may be computed the same way without any more programming effort. Note that the gradients w.r.t. $M$ may also be computed with the closed form formula $\nabla_M \sigma = u_1 v_1^\top$ where $u_1$ and $v_1$ are respectively the left and right singular vector of $M$ associated to the singular value $\sigma$ [27]. The same algorithm may be straightforwardly iterated to compute the $k$-largest singular values.

**Other layers.** Most activation functions such as ReLU, Leaky ReLU, SoftPlus, Tanh, Sigmoid, ArcTan or Softsign, as well as max-pooling, have a Lipschitz constant equal to $1$. Other common neural network layers such as dropout, batch normalization and other pooling methods all have simple and explicit Lipschitz constants. We refer the reader to e.g. [28] for more information on this subject.

## 6 Sequential neural networks

Despite its generality, AutoLip may be subject to large errors due to the multiplication of smaller errors at each iteration of the algorithm. In this section, we improve on the AutoLip upper bound by a more refined analysis of deep learning architectures in the case of MLPs. More specifically, the Lipschitz constant of MLPs have an explicit formula using Theorem 1 and the chain rule:

$$L(f_{MLP}) = \sup_{x \in \mathbb{R}^n} \|M_K \operatorname{diag}(g'_{K-1}(\theta_{K-1})) M_{k-1} ... M_2 \operatorname{diag}(g'_1(\theta_1)) M_1\|_2, \tag{6}$$

where $\theta_k = T_k \circ \rho_{k-1} \circ \cdots \circ \rho_1 \circ T_1(x)$ is the intermediate output after $k$ linear layers.

Considering Proposition 1 and Eq. (6), the equality $\hat{L}_{AL} = L(f_{MLP})$ only takes place if all activation layers $\operatorname{diag}(g'_k(\theta_k))$ map the first singular vector of $M_k$ to the first singular vector of $M_{k+1}$ by Cauchy-Schwarz inequality. However, differential operators of activation layers, being diagonal matrices, can only have a limited effect on input vectors, and in practice, first singular vectors will tend to misalign, leading to a drop in the Lipschitz constant of the MLP. This is the intuition behind *SeqLip*, an improved algorithm for Lipschitz constant estimation for MLPs.

## 6.1 SeqLip, an improved algorithm for MLPs

In Eq. (6), the diagonal matrices $\mathrm{diag}(g'_{K-1}(\theta_{K-1}))$ are difficult to evaluate, as they may depend on the input value $x$ and previous layers. Fortunately, as stated in Section 5, most major activation functions are 1-Lipschitz. More specifically, these activation functions have a derivative $g'_k(x) \in [0, 1]$. Hence, we may replace the supremum on the input vector $x$ by a supremum over all possible values:

$$L(f_{MLP}) \leq \max_{\forall i,\ \sigma_i \in [0,1]^{n_i}} \|M_K \, \mathrm{diag}(\sigma_{K-1}) \cdots \mathrm{diag}(\sigma_1) M_1\|_2, \tag{7}$$

where $\sigma_i$ corresponds to all possible derivatives of the activation gate. Solving the right hand side of Eq. (7) is still a hard problem, and the high dimensionality of the search space $\sigma \in [0,1]^{\sum_{i=1}^{K} n_i}$ makes purely combinatorial approaches prohibitive even for small neural networks. In order to decrease the complexity of the problem, we split the operator norm in $K-1$ parts using the SVD decomposition of each matrix $M_i = U_i \Sigma_i V_i^\top$ and the submultiplicativity of the operator norm:

$$L(f_{MLP}) \leq \max_{\forall i,\ \sigma_i \in [0,1]^{n_i}} \|\Sigma_K V_K^\top \, \mathrm{diag}(\sigma_K) U_{K-1} \Sigma_{K-1} V_{K-1}^\top \, \mathrm{diag}(\sigma_{K-1}) \ldots \mathrm{diag}(\sigma_1) U_1 \Sigma_1\|_2,$$

$$\leq \prod_{i=1}^{K-1} \max_{\sigma_i \in [0,1]^{n_i}} \left\| \widetilde{\Sigma}_{i+1} V_{i+1}^\top \, \mathrm{diag}(\sigma_{i+1}) U_i \widetilde{\Sigma}_i \right\|_2,$$

where $\widetilde{\Sigma}_i = \Sigma_i$ if $i \in \{1, K\}$ and $\widetilde{\Sigma}_i = \Sigma_i^{1/2}$ otherwise. Each activation layer can now be solved independently, leading to the *SeqLip* upper bound:

$$\hat{L}_{SL} = \prod_{i=1}^{K-1} \max_{\sigma_i \in [0,1]^{n_i}} \left\| \widetilde{\Sigma}_{i+1} V_{i+1}^\top \, \mathrm{diag}(\sigma_{i+1}) U_i \widetilde{\Sigma}_i \right\|_2. \tag{8}$$

When the activation layers are ReLU and the inner layers are small ($n_i \leq 20$), the gradients are $g'_k \in \{0, 1\}$ and we may explore the entire search space $\sigma_i \in \{0, 1\}^{n_i}$ using a brute force combinatorial approach. Otherwise, a gradient ascent may be used by computing gradients via the power method described in Alg. 2. In our experiments, we call this heuristic *Greedy SeqLip*, and verified that the incurred error is at most 1% whenever the exact optimum is computable. Finally, when the dimension of the layer is too large to compute a whole SVD, we perform a low rank-approximation of the matrix $M_i$ by retaining the first $E$ eigenvectors ($E = 200$ in our experiments).

## 6.2 Theoretical analysis of SeqLip

In order to better understand how SeqLip may improve on AutoLip, we now consider a simple setting in which all linear layers have a large difference between their first and second singular values. For simplicity, we also assume that activation functions have a derivative $g'_k(x) \in [0, 1]$, although the following results easily generalize as long as the derivative remains bounded. Then, the following theorem holds.

**Theorem 3.** *Let $M_k$ be the matrix associated to the $k$-th linear layer, $u_k$ (resp. $v_k$) its first left (resp. right) singular vector, and $r_k = s_{k,2}/s_{k,1}$ the ratio between its second and first singular values. Then, we have*

$$\hat{L}_{SL} \leq \hat{L}_{AL} \prod_{k=1}^{K-1} \sqrt{(1 - r_k - r_{k+1}) \max_{\sigma \in [0,1]^{n_k}} \langle \sigma \cdot v_{k+1}, u_k \rangle^2 + r_k + r_{k+1} + r_k r_{k+1}}.$$

Note that $\max_{\sigma \in [0,1]^{n_k}} \langle \sigma \cdot v_{k+1}, u_k \rangle^2 \leq 1$ and, when the ratios $r_k$ are negligible, then

$$\hat{L}_{SL} \leq \hat{L}_{AL} \prod_{k=1}^{K-1} \max_{\sigma \in [0,1]^{n_k}} |\langle \sigma \cdot v_{k+1}, u_k \rangle|. \tag{9}$$

Intuitively, each activation layer may align $u_k$ to $v_{k+1}$ only to a certain extent. Moreover, when the two singular vectors $u_k$ and $v_{k+1}$ are not too similar, this quantity can be substantially smaller than 1. To illustrate this idea, we now show that $\max_{\sigma \in [0,1]^{n_k}} |\langle \sigma \cdot v_{k+1}, u_k \rangle|$ is of the order of $1/\pi$ if the two vectors are randomly chosen on the unit sphere.

**Lemma 2.** *Let $x \geq 0$ and $u, v \in \mathbb{R}^n$ be two independent random vectors taken uniformly on the unit sphere $\mathbb{S}^{n-1} = \{x \in \mathbb{R}^n : \|x\|_2 = 1\}$. Then we have*

$$\max_{\sigma \in [0,1]^n} |\langle \sigma \cdot u, v \rangle| \xrightarrow[n \to +\infty]{} \frac{1}{\pi} \quad \textit{almost surely.}$$

Intuitively, when the ratios between the second and first singular values are sufficiently small, each activation layer decreases the Lipschitz constant by a factor $1/\pi$ and

$$\hat{L}_{SL} \approx \frac{\hat{L}_{AL}}{\pi^{K-1}}. \tag{10}$$

For example, for $K = 5$ linear layers, we have $\pi^{K-1} \approx 100$ and a large improvement may be expected for SeqLip compared to AutoLip. Of course, in a more realistic setting, the eigenvectors of different layers are not independent and, more importantly, the ratio between second and first eigenvalues may not be sufficiently small. However, this simple setting provides us with the best improvement one can hope for, and our experiments in Section 7 shows that at least part of the suboptimality of AutoLip is due to the misalignment of eigenvectors.

## 7 Experimentations

As stated in Theorem 2, computing the Lipschitz constant is an **NP**-hard problem. However, in low dimension (e.g. $d \leq 5$), optimizing the problem in Eq. (1) can be performed efficiently using a simple grid search. This will provide a baseline to compare the different estimation algorithms. In high dimension, grid search is intractable and we consider several other estimation methods: (1) grid search for Eq. (1), (2) simulated annealing for Eq. (1), (3) product of Frobenius norms of linear layers [13], (4) product of spectral norms [13] (equivalent to AutoLip in the case of MLPs). Note that, for MLPs with ReLU activations, first order optimization methods such as SGD are not usable because the function to optimize in Eq. (1) is piecewise constant. Methods (1) and (2) return lower bounds while (3) and (4) return upper bounds on the Lipschitz constant.

**Ideal scenario.** We first show the improvement of SeqLip over AutoLip in an ideal setting where inner layers have a low eigenvalue ratio $r_k$ and uncorrelated leading eigenvectors. To do so, we construct an MLP with weight matrices $M_i = U_i \operatorname{diag}(\lambda) V_i^\top$ such that $U_i, V_i$ are random orthogonal matrices and $\lambda_1 = 1, \lambda_{i>1} = r$ where $r \in [0, 1]$ is the ratio between first and second eigenvalue. Figure 2 shows the decrease of SeqLip as the number of layers of the MLP increases (each layer has 100 neurons). The theoretical limit is tight for small eigenvalue ratio. Note that the AutoLip upper bound is always 1 as, by construction of the network, all layers have a spectral radius equal to one.

**MLP.** We construct a 2-dimensional dataset from a Gaussian Process with RBF Kernel with mean 0 and variance 1. We use 15000 generated points as a synthetic dataset. An example of such a dataset may be seen in Figure 3. We train MLPs of several depths with 20 neurons at each layer, on the synthetic dataset with MSE loss and ReLU activations. Note that in all simulations, the greedy SeqLip algorithm is within a $0.01\%$ error compared to SeqLip, which justify its usage in higher dimension.

| # layers | Upper bounds | | | | Lower bounds | | |
|---|---|---|---|---|---|---|---|
| | Frobenius | AutoLip | SeqLip | Greedy SeqLip | Dataset | Annealing | Grid Search |
| 4 | 648.2 | 33.04 | 21.47 | 21.47 | 4.36 | 4.55 | 6.56 |
| 5 | 4283.1 | 134.4 | 72.87 | 72.87 | 6.77 | 5.8 | 7.1 |
| 7 | 22341 | 294.6 | 130.2 | 130.2 | 5.4 | 5.27 | 6.51 |
| 10 | 7343800 | 19248.2 | 2463.44 | 2463.36 | 10.04 | 5.77 | 17.1 |

Figure 4: AutoLip and SeqLip for MLPs of various size.

First, since the dimension is low ($d = 2$), grid search returns a very good approximation of the Lipschitz constant, while simulated annealing is suboptimal, probably due to the presence of local

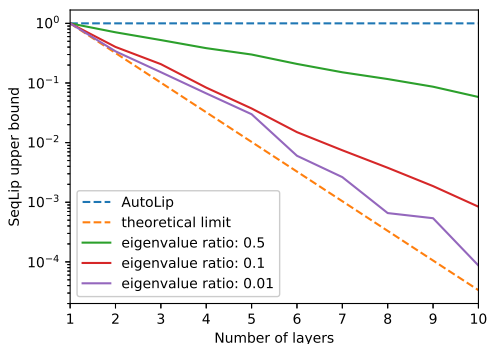
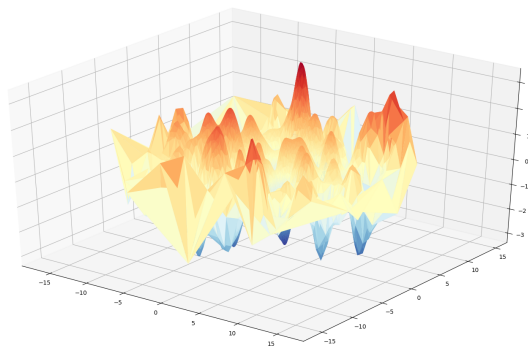

Figure 2: SeqLip in the ideal scenario.　　Figure 3: Synthetic function used to train MLPs.

maxima. For upper bounds, SeqLip outperforms its competitors reducing the gap between upper bounds and, in this case, the true Lipschitz constant computed using grid search.

**CNN.** We construct simple CNNs with increasing number of layers that we train independently on the MNIST dataset [29].The details of the structure of the CNNs are given in the supplementary material. SeqLip improves by a factor of $5$ the upper bound given by AutoLip for the CNN with $10$ layers. Note that the lower bounds obtained with simulated annealing is probably too low, as shown in the previous experiments.

| # layers | Upper bounds | | | Lower bounds | |
|---|---|---|---|---|---|
| | AutoLip | Greedy SeqLip | Ratio | Dataset | Annealing |
| 4 | 174 | 86 | 2 | 12.64 | 25.5 |
| 5 | 790.1 | 335 | 2.4 | 16.79 | 22.2 |
| 7 | 12141 | 3629 | 3.3 | 31.22 | 43.6 |
| 10 | $4.5 \cdot 10^6$ | $8.2 \cdot 10^5$ | 5.4 | 38.26 | 107.8 |

Figure 5: AutoLip and SeqLip for MNIST-CNNs of various size.

**AlexNet.** AlexNet [1] is one of the first successes of deep learning in computer vision. The AutoLip algorithm finds that the Lipschitz constant is upper bounded by $3.62 \times 10^7$ which remains extremely large and probably well above the true Lipschitz constant. As for the experiment on a CNN, we use the 200 highest singular values of each linear layer for Greedy SeqLip. We obtain $5.45 \times 10^6$ as an upper bound approximation, which remains large despite its 6 fold improvement over AutoLip. Note that we do not get the same results as [9, Section 4.3] as we did not use the same weights.

## 8 Conclusion

In this paper, we studied the Lispchitz regularity of neural networks. We first showed that exact computation of the Lipschitz constant is an **NP**-hard problem. We then provided a generic upper bound called *AutoLip* for the Lipschitz constant of any automatically differentiable function. In doing so, we introduced an algorithm to compute singular values of affine operators such as convolution in a very efficient way using *autograd* mechanism. We finally proposed a refinement of the previous method for MLPs called *SeqLip* and showed how this algorithm can improve on AutoLip theoretically and in applications, sometimes improving up to a factor of $8$ the AutoLip upper bound. While the AutoLip and SeqLip upper bounds remain extremely large for neural networks of the computer vision literature (e.g. AlexNet, see Section 7), it is yet an open question to know if these values are close to the true Lipschitz constant or substantially overestimating it.

**Acknowledgements**

The authors thank the whole team at Huawei Paris and in particular Igor Colin, Moez Draief, Sylvain Robbiano and Albert Thomas for useful discussions and feedback.

## Footnotes

[1]The code used in this paper is available at `https://github.com/avirmaux/lipEstimation`.

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
