[Supplementary Material]

# Lipschitz regularity of deep neural networks: analysis and efficient estimation

# SUPPLEMENTARY MATERIAL

**Kevin Scaman**
Huawei Noah's Ark Lab
`kevin.scaman@huawei.com`

**Aladin Virmaux**
Huawei Noah's Ark Lab
`aladin.virmaux@huawei.com`

## Abstract

This supplementary material contains proof of the theorems of the submission "Lipschitz regularity of deep neural networks: analysis and efficient estimation", as well as more details on the parameters used for the experiments.

## 1   Proof of Theorem 2

We reduce the problem of maximizing a quadratic convex function on a hypercube to **LIP-CST**. Start from the following **NP**-hard problem [1, Quadratic Optimization, Section 4]:

$$
\begin{aligned}
\underset{\sigma}{\text{maximize}} \quad & \sum_i (h_i^\top \sigma)^2 = \sigma^\top H \sigma \\
\text{s.\,t.} \quad & \forall k,\ 0 \le \sigma_k \le 1\,,
\end{aligned}
\tag{1}
$$

where $H = \sum_i h_i h_i^\top$ is a positive semi-definite matrix with full rank. Let's note

$$
M_1 = \left( \begin{array}{c|c|c|c} h_1 & h_2 & \cdots & h_n \end{array} \right), \quad
M_2 = \left( \begin{array}{c|c} \begin{matrix} 1 \\ \vdots \\ 1 \end{matrix} & 0 \end{array} \right)
$$

so that we have

$$
M_2 \operatorname{diag}(\sigma) M_1 = \left( \begin{array}{c|c} \begin{matrix} h_1^\top \sigma \\ \vdots \\ h_n^\top \sigma \end{matrix} & 0 \end{array} \right).
$$

The spectral norm of this 1-rank matrix is $\sum_i (h_i^\top \sigma)^2$. We proved that Eq. (1) is equivalent to the following optimization problem

$$
\begin{aligned}
\underset{\sigma}{\text{maximize}} \quad & \|M_2 \operatorname{diag}(\sigma) M_1\|_2^2 \\
\text{s.\,t.} \quad & \sigma \in [0,1]^n\,.
\end{aligned}
\tag{2}
$$

We recover the exact formulation of Section 6 Eq. (6) for a 2-layer MLP (the reader can verify there is no recursive loop). Because $H$ is full rank, $M_1$ is surjective and all $\sigma$ are admissible values for $g_i'(x)$ which is the equality case. Finally, ReLU activation units take their derivative within $\{0, 1\}$ and Eq. (2) is its relaxed optimization problem, that has the same optimum points.

## 2 Proof of Theorem 3

Consider a single factor $\left\|\widetilde{\Sigma} V \operatorname{diag}(\sigma) U^\top \widetilde{\Sigma}'\right\|_2$ with $V$ and $U$ unitary matrices and $\widetilde{\Sigma}$ (resp. $\widetilde{\Sigma}'$) is diagonal with eigenvalues $(s_k)_k$ (resp. $(s'_j)_j$) in decreasing order along the diagonal. Decompose the eigenvalue matrices as $\widetilde{\Sigma} = s_1 E_{11} + D$ and $\widetilde{\Sigma}' = s'_1 E'_{11} + D'$, by orthogonality we can write

$$\left\|\widetilde{\Sigma} V \operatorname{diag}(\sigma) U^\top \widetilde{\Sigma}'\right\|_2^2 \leq \left\| s_1 E_{11} V \operatorname{diag}(\sigma) U^\top E'_{11} s'_1 \right. \tag{3}$$

$$+ s_1 E_{11} V_i \operatorname{diag}(\sigma) U^\top D'$$

$$\left. + D V \operatorname{diag}(\sigma) U^\top E'_{11} s'_1 \right\|_2^2$$

$$+ \left\| D V \operatorname{diag}(\sigma) U^\top D' \right\|_2^2 . \tag{4}$$

First we can bound $(4) \leq (s_2 s'_2)^2$. For (3) denote $v_k$ (resp. $u_k$) the $k$-th column of $V$ (resp. of $U$). It follows that

$$(3) \leq (s_1 s'_1)^2 \langle v_1, \sigma \cdot u_1 \rangle^2 + \sum_{j>1} (s_1 s'_j)^2 \langle v_1, \sigma \cdot u_j \rangle^2 + \sum_{k>1} (s_k s'_1)^2 \langle v_k, \sigma \cdot u_1 \rangle^2 .$$

The columns $(v_k)_k$ of $V$ form an orthonormal basis so we have

$$\sum_{k>1} \langle v_k, \sigma \cdot u_1 \rangle^2 = \|\sigma \cdot u_1\|^2 - \langle v_1, \sigma \cdot u_1 \rangle^2 ,$$

and we deduce a similar equality for $\sum_{j>1} \langle v_1, \sigma \cdot u_j \rangle^2$. Using $s_k \leq s_2$ for $k > 1$ we finally obtain

$$(3) \leq (s_1 s'_1)^2 \left( \langle v_1, \sigma \cdot u_1 \rangle^2 (1 - \widetilde{r}_1 - \widetilde{r}_2) + \widetilde{r}_1 + \widetilde{r}_2 \right) ,$$

with $\widetilde{r}_1 = (\frac{s_2}{s_1})^2$ and $\widetilde{r}_2 = (\frac{s'_2}{s'_1})^2$. In conclusion we proved the following inequality:

$$\left\|\widetilde{\Sigma} V \operatorname{diag}(\sigma) U^\top \widetilde{\Sigma}'\right\|_2^2 \leq (s_1 s'_1)^2 \left( (1 - \widetilde{r}_1 - \widetilde{r}_2) \langle v_1, \sigma \cdot u_1 \rangle^2 + \widetilde{r}_1 + \widetilde{r}_2 + \widetilde{r}_1 \widetilde{r}_2 \right) .$$

The Lipschitz upper bound given by AutoLip of $\left\|\widetilde{\Sigma}_1 V \operatorname{diag}(\sigma) U^\top \widetilde{\Sigma}_2\right\|_2$ is $s_1 s'_1$. For the middle layers, we have $\widetilde{\Sigma} = \Sigma^{1/2}$, and the inequality still holds for the first and last layer due to $\widetilde{r}_i \leq \frac{s_2}{s_1}$; taking the maximum for $\sigma$ leads to the theorem.

## 3 Proof of Lemma 2

Let $U, V \sim \mathcal{N}(0, I_n)$ be two independent $n$-dimensional Gaussian random vectors. Then, $u = U/\|U\|_2$ and $v = V/\|V\|_2$ are uniform on the unit sphere $\mathcal{S}^{n-1}$, and

$$\max_{\sigma \in [0,1]^n} |\langle \sigma \cdot u, v \rangle| = \max_{\sigma \in [0,1]^n} \left| \sum_{i=1}^n \sigma_i u_i v_i \right|$$

$$= \max \left\{ \sum_{i=1}^n (u_i v_i)^+, \sum_{i=1}^n (u_i v_i)^- \right\} , \tag{5}$$

where $x^+ = \max\{0, x\}$ and $x^- = \max\{0, -x\}$ are respectively the positive and negative parts of $x$. Note that $\sum_{i=1}^n (u_i v_i)^+$ and $\sum_{i=1}^n (u_i v_i)^-$ have the same law, since the distribution of $u$ and $v$ is symmetric w.r.t. the coordinate axes. Moreover, we may rewrite

$$\sum_{i=1}^n (u_i v_i)^+ = \frac{\frac{1}{n} \sum_{i=1}^n (U_i V_i)^+}{\sqrt{\frac{1}{n} \sum_{i=1}^n U_i^2} \sqrt{\frac{1}{n} \sum_{i=1}^n V_i^2}}, \tag{6}$$

and each term converges almost surely to its expectation due to the strong law of large numbers. Finally, noting that $\mathbb{E}\left[U_i^2\right] = \mathbb{E}\left[V_i^2\right] = 1$ and

$$\mathbb{E}\left[(U_i V_i)^+\right] = \frac{1}{2} \mathbb{E}\left[|U_i V_i|\right] = \frac{1}{2} \mathbb{E}\left[|U_i|\right] \mathbb{E}\left[|V_i|\right] = \frac{1}{\pi}, \tag{7}$$

leads to the desired result.

## 4  Convolutional Neural Network of Section 7

For each model of depth $n$, convolution except the last one are followed by a ReLU activation unit.

| # Layer | Layer | # channels out | kernel | stride | padding |
|---------|-------|----------------|--------|--------|---------|
| 1 | Conv2D + bias | 32 | $(5,5)$ | 2 | 0 |
| 2 | Conv2D + bias | 64 | $(3,3)$ | 2 | 0 |
| 3 | Conv2D + bias | 64 | $(3,3)$ | 1 | 1 |
| $\vdots$ | $\vdots$ | $\vdots$ | $\vdots$ | $\vdots$ | $\vdots$ |
| $\vdots$ | Conv2D + bias | 64 | $(3,3)$ | 1 | 1 |
| $\vdots$ | $\vdots$ | $\vdots$ | $\vdots$ | $\vdots$ | $\vdots$ |
| $n-1$ | Conv2D + bias | 128 | $(3,3)$ | 2 | 0 |
| $n$ | Conv2D + bias | 10 | $(2,2)$ | 1 | 0 |

## References

[1] Reiner Horst and Panos M Pardalos. *Handbook of global optimization*, volume 2. Springer Science & Business Media, 2013.