[Reviews · NeurIPS 2018]

Reviewer 1



The paper studies the problem of estimating the Lipschitz constant of a mapping realized by a trained neural network. Several distinct contributions are made: (i) It is shown that exactly computing the Lipschitz constant of a two layer MLP (Multi Layer Preceptron) with ReLU activation is NP-hard (worst-case result). (ii) A straightforward algorithm for bounding the Lipschitz constant of a computational auto-differentiable graph is given. The algorithm, named AutoLip, is a generalization of multiplying spectral norms of Jacobians for a sequence of mappings, and hence usually provides pessimistic bounds. (iii) A technique for efficiently applying power method to a high-dimensional affine transformation is given. This technique assumes efficient implementation of the mapping in an auto-differentiable graph. It allows estimating the leading singular value of the transformation, which is equal to the Lipschitz constant. (iv) For the case of MLP, an algorithm named SeqLip is presented, which is less naive, and tighter than AutoLip. The degree to which it is tighter is theoretically analyzed under simplifying assumptions. The paper concludes with experiments comparing different estimations of the Lipschitz constant for several trained networks. In my opinion, the problem studied in this paper is important, not only for robustness to input perturbations (motivation given in the text), but also for improving recent generalization bounds that depend on Lipschitz constant. I advise the authors to mention this latter point in the text. In terms of quality, the paper is well written, and its contributions are interesting. I did not verify the proofs in supplementary material, but the results seem very reasonable and I believe they are correct. One point I find troubling is the connection, or lack thereof, to prior work -- absolutely nothing is said about existing literature. I kindly ask the authors to include such reference in their rebuttal, and subsequently add it to the manuscript. To my knowledge contributions (i)-(iv) above are novel. Assuming this is the case, I recommend accepting the paper. Points which I believe can improve this work: * The text covers multiple ideas, many of which are in my opinion distinct, and vary in their importance. Of contributions (i)-(iv) above, I think (iv) is the most interesting one, and (iii), which is also interesting, can be presented as part of (iv). (i) and (ii) are less interesting (the first because it is a worst case analysis; the second because it is trivial). There is also some text beyond (i)-(iv) which I think adds almost nothing -- the discussion on Lipschitz constants of activation, pooling, dropout and batch norm layers. I believe the text would benefit from being more focused on the important parts. * It seems that the improvement of SeqLip over AutoLip is not that significant (one order of magnitude, while AutoLip is likely overpessimistic by much more than that). I think further refinement of SeqLip would be of great interest, and encourage the authors to pursue this goal, either in this work or in a follow-up. * The experiments are very minimal, which is a shame because they are insightful. I would add many more models to the evaluation, placing the results in supplementary material if needed. * I think there are some typos in the text (for example in Eq 11). This is not critical, because the reader can easily reproduce the derivations independently. ---- Edit following discussion period: I have read the authors' response, and in particular their reference to existing literature. I have also read the other reviews. While I agree with the criticism of R3 with regards to the experiments, in my opinion, the contribution of this work is theoretical -- it presents a technique (SeqLip) for bounding the Lipschitz constant of a MLP, going a step beyond the trivial routine of multiplying spectral norms. My stand on the paper remains -- I recommend its acceptance.

Reviewer 2



This paper studies the Lipschitz constant of deep neural networks, proving that an exact computation is NP-hard and providing an algorithm for computing an upper bound using automatic differentiation. The proposed method and its variants outperform baselines and provide the first bounds for some large-scale networks used in practice. The Lipschitz constant of deep neural networks is an important quantity characterizing sensitivity to small perturbations in the input, which may be relevant for understanding aspects of trainability, generalization, and robustness to adversarial examples. For these reasons I think the topic of this paper is salient and the tools and ideas developed here will be useful and interesting to the deep learning community. Although I did not review the proofs in detail, the analysis appears sound and the paper is well-written. One important question is how important this type of worst-case analysis is for practice, i.e. do typical real-world high-dimensional data distributions typically include datapoints for which the maximum singular value of the Jacobian is close to the Lipschitz constant? I could imagine that typically these worst-case inputs are very unnatural, and that datapoints from the actual data distribution end up exhibiting close to average-case behavior. It might be worth including a discussion of these considerations. As a minor point, I'd encourage sampling the synthetic function in Figure 2 with a finer mesh to produce a smoother figure. Overall, I think this is a good paper that would be appreciated by the NIPS community and vote for acceptance.

Reviewer 3



In this paper, the authors presented two methods to estimate the Lipschitz regularity of deep neural networks to understand their sensitivity to perturbation of inputs. A generic algorithm AutoLip is proposed which is applicable to any automatically differentiable function. An improved algorithm called SeqLip is proposed for sequential neural networks. Some experimental results are given to illustrate the effectiveness of these methods. (1) While it is claimed that AutoLip is applicable to any automatically differentiable function, it seems that its effectiveness is tested only on the specifical sequential neural networks. It would be helpful to present experiments with more general neural networks. (2) The algorithm SeqLip requires SVD decomposition for each matrix in each layer. I am afraid that the computation can be expensive for large $n_i$. (3) In the first line of eq (11), it seems that the right-hand side should be $$ \|\Sigma_1U_1^\top\diag(\sigma_1)V_2\Sigma_2U_2^\top\diag(\sigma_2)\cdots V_{k-1}\Sigma_{k-1}U_{k-1}^\top\diag(\sigma_{k-1})V_K\Sigma_k\|_2. $$ I am not sure whether this is equal to the one mentioned in the paper. (4) The experimental results are not quite convincing. For the experiment with MLP, the dimension is too small. For the experiment with CNN, the depth is a bit small. For the experiment with AlexNet, the Lipschitz constant obtained by Greedy SeqLip is still too large. (5) I did not understand the meaning of ``a brute force combinational approach'' in Section 6.1. As far as I can see, this involves no combinational optimization problems. Minor Comments: (1) In eq (11), (12), $k$ should be $K$ (2) In conclusion, it is stated that SeqLip outperforms AutoLip by a factor of $10$. However, I can only see an improvement up to a factor of $6$ in the experiments. --------------------------- After rebuttal and discussions: I agree with Reviewer #1 and Reviewer #2 that the proposed SeqLip is interesting for bounding the Lipschitz constant of a MLP. I vote for its acceptance.